# ONE-SHOT WORLD MODELS USING A TRANSFORMER TRAINED ON A SYNTHETIC PRIOR

## ABSTRACT

A World Model is a compressed spatial and temporal representation of a real world environment that allows one to train an agent or execute planning methods. However, world models are typically trained on observations from the real world environment, and they usually do not enable learning policies for other real environments. We propose *One-Shot World Model* (OSWM), a transformer world model that is learned in an in-context learning fashion from purely synthetic data sampled from a prior distribution. Our prior is composed of multiple randomly initialized neural networks, where each network models the dynamics of each state and reward dimension of a desired target environment. We adopt the supervised learning procedure of Prior-Fitted Networks by masking next-state and reward at random context positions and query OSWM to make probabilistic predictions based on the remaining transition context. During inference time, OSWM is able to quickly adapt to the dynamics of a simple grid world, as well as the CartPole gym and a custom control environment by providing 1k transition steps as context and is then able to successfully train environment-solving agent policies. However, transferring to more complex environments remains a challenge, currently. Despite these limitations, we see this work as an important stepping-stone in the pursuit of learning world models purely from synthetic data.

## 1 INTRODUCTION

World models have emerged as a powerful approach for creating compressed spatial and temporal representations of real-world environments, enabling efficient agent training and planning in reinforcement learning (RL) tasks (Ha & Schmidhuber, 2018; Kaiser et al., 2019; Hafner et al., 2023; Wu et al., 2022). These models have shown significant promise in improving sample efficiency and performance across various RL domains. For instance, SimPLe (Kaiser et al., 2019) demonstrated strong results on Atari games by using a learned dynamics model to generate simulated data. More recently, transformer-based world models have pushed the boundaries of sample efficiency and performance. TWM (Robine et al., 2023) utilized a Transformer-XL architecture to surpass other methods on the Atari 100k benchmark Kaiser et al. (2019), while IRIS (Micheli et al., 2023) and STORM (Zhang et al., 2023) achieved human-level performance using GPT-style transformers. However, these approaches typically require training on observations from the target environment, which can be time-consuming and impractical in many real-world scenarios. Moreover, traditional world models often lack the ability to generalize across different environments, limiting their applicability in diverse RL tasks. The challenge of transferring learned dynamics efficiently to new environments remains a significant hurdle in the field of model-based RL.

To address these challenges, we explore the potential of training world models with in-context learning using purely synthetic data. We propose the One-Shot World Model (OSWM), a transformer-based approach that learns a world model from a synthetic prior distribution. Our method draws inspiration from Prior-Fitted Networks (Müller et al., 2022) and leverages in-context learning to adapt to new environments with minimal real-world interactions. By training on a diverse synthetic prior, OSWM aims to capture a wide range of environment dynamics, potentially enabling rapid adaptation to various RL tasks. We release our code under `https://anonymous.4open.science/r/PFN-SE-7ABB/` and our contributions can be summarized as follows:

Figure 1: OSWM is trained on synthetic data sampled from a prior distribution of randomly initialized, untrained neural networks that mimic RL environments (left). Given a sequence of synthetic interactions, OSWM is optimized by predicting future dynamics at random cut-offs (center). RL agents can then be trained on OSWM to solve simple real environments given a context.

- We explore training world models with synthetic data (transition sequences) sampled from a novel synthetic prior distribution based on randomly initialized and untrained neural networks. We train the transformer world model purely on data sampled from this synthetic prior in an in-context learning fashion, predicting future dynamics and rewards based on previous state and action sequences.

- We demonstrate that our model, One-Shot World Model (OSWM), is capable of adapting to the dynamics of unseen environments in a one-shot manner by only providing 1,000 randomly sampled transitions as context.

- Although OSWM adaptability is still limited to very simple environments, we show that training world models on such a synthetic prior surprisingly allows for the training of RL agents that solve the GridWorld, CartPole gym and a custom control environment.

- We investigate OSWM's limitations and analyze strategies for prior construction and the relevance of context sampling, providing insights for future improvements in this direction.

## 2 RELATED WORK

**World Models and Model-Based Reinforcement Learning** Classical RL often suffers from sample inefficiency, as it requires many interactions with the environment. Model-Based Reinforcement Learning (MBRL) mitigates this by learning environment dynamics, allowing agents to train using simulated data. For example, Ha & Schmidhuber (2018) proposed World Models, which use generative neural networks to encode compact spatial-temporal representations, optimizing RL efficiency. MuZero (Schrittwieser et al., 2020) advanced MBRL by learning both environment dynamics and reward functions, which proved highly effective across board games. Dreamer (Hafner et al., 2020; 2021; 2023) applied learned world models across diverse domains, including real-world robotics (Wu et al., 2022). More recently, TD-MPC2 (Hansen et al., 2024) demonstrated scalability and robustness in continuous control tasks. Transformer-based models have also become prominent, with TransDreamer (Chen et al., 2022) or TWM (Robine et al., 2023) that excelled in sample efficiency on Atari 100k. Other Transformer-based approaches such as IRIS (Micheli et al., 2023) or STORM (Zhang et al., 2023) achieved over 100% human performance on Atari 100k with GPT-style training. However, Most if not all methods are trained on the target environment and utilize the attention mechanism to attend to previous parts of the roll-out.

Despite these successes, transferring learned dynamics across environments remains a significant hurdle in the field of MBRL. Augmented World Models (Ball et al., 2021) tackle environmental dynamics changes by learning a world model from offline data. During training, they provide predicted dynamics and possible changes as latent context, helping agents generalize to new variations. Similarly, Evans et al. (2022) uses transformers or RNNs to encode environment parameterization into a latent space, enabling a world model robust to variations like friction or object mass changes.

**Synthetic Data and Priors and RL** Synthetic data plays a crucial role in RL, particularly in methods that transfer knowledge from simulation to real environments, known as *sim2real*. Domain randomization (Tobin et al., 2017), which varies simulation settings like lighting and object shapes, enhances generalization and improves the transfer from simulation to the real world. Pretraining

with synthetic data has also gained prominence. For example, Wang et al. (2024) pretrains the Decision Transformer using synthetic Markov chain data, outperforming pretraining with natural text (e.g., DPT trained on Wikipedia (Lee et al., 2023)) in both performance and sample efficiency. Other techniques include training on synthetic reward distributions to allow zero-shot transfer to new tasks (Frans et al., 2024), while TDM (Schubert et al., 2023) demonstrates strong few-shot and zero-shot generalization across procedural control environments. UniSim (Yang et al., 2023) uses internet-scale data to train realistic robotic control models, enabling more efficient RL training. A meta-learning approach trains Synthetic Environments (Ferreira et al., 2021) for RL that serve as proxies for a target environment, providing only synthetic environment dynamics. These synthetic dynamics allow RL agents to significantly reduce the number of interactions needed during training. Lastly, Prior Fitted Networks (PFNs) (Müller et al., 2022) utilize synthetic priors for supervised learning, with its adaptation to tabular data, TabPFN (Hollmann et al., 2023), achieving state-of-the-art results while significantly speeding up inference.

Unlike previous approaches that depend on real-world observations or extensive training in target environments, we introduce a new approach that trains a transformer world model entirely on synthetic data sampled from a prior distribution which is based much further away from reality as it based on randomly initiliazed neural networks. Using the Prior-Fitted Networks paradigm, OSWM employs in-context learning to adapt to new environments with just a simple context sequence.

## 3 METHOD

Let $x_t = \left[ s_t^{1:d_s}, a_t^{1:d_a} \right]$ denote the concatenated state-action vector (or *input*) at time step $t$, where $s_t^{1:d_s}$ represents the state and $a_t^{1:d_a}$ represents the action, with $d_s$ and $d_a$ being the dimensionalities of the state and action, respectively. Similarly, let $y_t = \left[ s_{t+1}^{1:d_s}, r_{t+1} \right]$ represent the next state and reward vector (or *target*). The sequences of these vectors, $\{x_1, \ldots, x_T\}$ and $\{y_1, \ldots, y_T\}$, are summarized as $X_{1:T}$ and $Y_{1:T}$, respectively. To ensure consistent input sizes across varying environments, padding is applied: $x_t = [s_t^1, ..., s_t^{d_s}, pad_s, a_t^1, ..., a_t^{d_a}, pad_a]$, where $pad_s$ and $pad_a$ are zero vectors used to match the maximum state and action dimensions across environments. The same padding scheme is applied to $y_t$. The OSWM is trained on synthetic batches $(X_{1:T}, Y_{1:T})$ sampled from a prior distribution $P_{RL}$. At randomly sampled cut-off positions, the synthetic batches are divided into context and target data and the model is trained to predict the target data given the context, which we visualized in Fig. 1 (center).

At inference, OSWM adapts to a new environment using a few context samples $(X_{1:T-1}, Y_{1:T-1})$ collected from the real environment, i.e. the target environment (see Fig. 1). This context consist of state-action transitions and their corresponding rewards, which provide information about the dynamics of the real environment. To ensure sufficient coverage of the target environment, multiple transitions are collected, often spanning several episodes. We typically collect 1,000 transitions from random rollouts, though the collection process can be performed using any policy, ranging from random to expert-driven actions. We analyze the role of context generation on the predictive performance of the model in Section 4.3.

Once the context is collected, OSWM predicts the next state and reward $(s_{t+1}, r_{t+1})$ given the current state-action pair $(s_t, a_t)$ and the prior context. The OSWM acts as a learned simulator, enabling RL agents to interact with predicted dynamics and learning by standard RL algorithms. Note, that the OSWM is initialized by sampling an initial state from the real environment at inference time. Both inputs $X_{1:T}$ and targets $Y_{1:T}$ are normalized to zero mean and unit variance. OSWM predicts in this normalized space, and the predictions are projected back to the original value space using the mean and variance of the context data. Finally, we assume that the termination condition of the target environment is known, but we note that it could also be learned.

### 3.1 TRAINING THE ONE-SHOT WORLD MODEL (OSWM)

The OSWM is trained on synthetic data sampled from a prior distribution $P_{RL}$ (see Section 3.2), which is constructed to simulate the dynamics of various environments. The goal is to optimize the model for predicting the dynamics of unseen target environments based on initial interactions used by in-context learning. We describe the entire training procedure in Algorithm 1.

At first, the model weights $\theta$ are initialized randomly. During each training step, a batch of $(X_{1:T}, Y_{1:T}) \sim \mathcal{P}_{RL}$ is sampled, with each batch containing input and target sequences. A context size $eval$ is sampled from the interval $[k, T-1]$, where $k$ is the minimum context size used for the prediction (see Appendix C for more details about the sampling). The model is provided with $X_{1:eval}$ and $Y_{1:eval}$ to predict future targets $\hat{Y}_{eval+1:T}$ based on the remaining inputs $X_{eval+1:T}$. The training loss is computed using the mean-squared error (MSE) between the predicted and actual future transitions: $L = MSE(\hat{Y}_{eval+1:T}, Y_{eval+1:T})$.

---

**Algorithm 1** Training the OSWM with the synthetic prior

---

    Initialize $\theta$                                                              ▷ Initialize OSWM's parameters
    **while** not finished **do**
        $X_{1:T}, Y_{1:T} \sim \mathcal{P}_{RL}$                        ▷ Sample batch from RL prior
        $eval \sim \mathcal{U}(k, T-1)$                       ▷ Sample $eval$ size
        $\hat{Y}_{eval\_pos+1:T} \leftarrow \mathcal{M}_\theta(X_{1:eval}, Y_{1:eval}, X_{eval+1:T})$    ▷ Predict dynamics with OSWM
        $L \leftarrow MSE(\hat{Y}_{eval+1:T}, Y_{eval+1:T})$             ▷ Calculate loss
        $\theta \leftarrow \theta - \alpha \nabla_\theta L$                       ▷ Update parameters
    **end while**
    **return** $\mathcal{M}_\theta$

---

### 3.2 Prior for Training OSWM

One of the core contributions of this method is the design of a prior that aims to mimic the properties of RL environments while incorporating stochasticity for diverse dynamics. The prior consists of two components: a neural network-based (NN) prior and a physics-based momentum prior. These two priors are combined, with the states produced by both the NN and momentum priors concatenated as input to the NN prior for further updates. This split allows the model to capture both complex, neural network-generated behaviors and simple, physics-driven interactions, like pendulum motion or velocity-position relations. In Figure 1 (left), we illustrate the mechanics of the NN prior, and below we describe both priors in more detail.

**Neural Network Prior** The NN prior generates dynamics using randomly initialized neural networks. Each state dimension $s_t^i$ is produced by a separate neural network $f_{\theta^i}^i$, which is randomly-initialized and untrained and takes as input the entire previous state $s_{t-1} = [s_{t-1}^1, ..., s_{t-1}^{d_s}]$ and action $a_t = [a_{t-1}^1, ..., a_t^{d_a}]$. The next state is computed as $s_t^i = f_{\theta^i}^i(s_{t-1}, a_{t-1})$. The networks consist of three linear layers, with random activations (ReLU, tanh, or sigmoid) after the first two layers, and a residual connection that aggregates the outputs of the first and second layers. This structure allows for complex dependencies between state dimensions and actions. To introduce variability, each NN-based state dimension is initialized with a random scale and offset. When the individual NN prior networks are reset, which occurs periodically after a pre-defined fixed interval, their initial state values $s_0$ are sampled from $\mathcal{U}(0, 1)$, and then scaled and offset according to the prior configuration (see Table 6 for the prior hyperparameters in the appendix), ensuring stochastic behavior across environments. This method allows the model to capture rich and diverse dynamics by introducing different dependencies between states and actions across dimensions.

**Momentum Prior** The momentum prior models physical interactions through two components: velocity and positional updates. Velocity is updated based on the action and gravity ($v_{t+1} = v_t + a_t \cdot \Delta t - g \cdot \Delta t$), while position is updated using the current velocity ($p_{t+1} = p_t + v_{t+1} \cdot \Delta t$). In this model, velocity $v_t$ and position $p_t$ are influenced by factors such as gravity and the current action, and the position updates rely on velocity. The initial position is sampled from $[0, 2\pi]$, and the initial velocity is sampled from $\mathcal{U}(-3, 3)$. This setup enables the model to simulate both linear and angular motion. Angular dynamics can incorporate gravity, and they are represented internally in radians, though the output can be sine, cosine, or radian values. The momentum prior values are concatenated with the NN prior values and fed into the NN prior networks for the subsequent transitions.

**Rewards and Invariance** The reward function follows a similar structure to the NN prior used for state dynamics but with different inputs, including the new state, action, and the previous state. This reflects how rewards in real RL environments are based on state transitions and action costs, such as penalizing high action magnitudes. The reward at time step $t$ can be expressed as: $r_{t+1} = g(s_{t+1}, a_t, s_t)$ where $g$ represents the reward model that takes the new state $s_{t+1}$, the action $a_t$, and, optionally, the previous state $s_t$ as inputs. To maintain flexibility, the reward is replaced by a constant reward of 1 with a probability of 0.5, a common approach in tasks like CartPole, where extending the episode is rewarded, or MountainCar, where faster completion is incentivized. To prevent the model from overfitting to the order of state-action dimensions, we shuffle both states and actions and apply identical permutations to $X_1$ and $Y_1$.

## 4 EXPERIMENTS

We first test the model's performance on various environments with the goal to provide an overview of the capabilities and limitations. We then describe how different prior components affect the predictions of OSWM and explore the impact of various context generation methods.

### 4.1 RESULTS FOR AGENT TRAINING

We evaluate the performance of OSWM by training an RL agent using the PPO algorithm (Schulman et al., 2017), as implemented in stable-baselines 3 (Raffin et al., 2021). We chose PPO because it can handle both discrete and continuous action spaces, making it well suited for the variety of environments in this study. We selected environments that provide a mix of discrete and continuous state and action spaces, allowing us to assess OSWM's performance across different types of RL challenges. The selected environments include two custom environments, GridWorld and SimpleEnv (see Appendix D for details), as well as CartPole-v0, MountainCar-v0, Pendulum-v1, and Reacher-v4 from the classic control gym suite.

In GridWorld, the agent navigates a discrete, 8x8 grid to reach a target location, receiving a positive reward for reaching the target and small penalties for each step, and the environment is considered solved when the agent consistently reaches the target efficiently. SimpleEnv involves moving a point along a 1D continuous line toward the center, with rewards negatively proportional to the distance from the center. CartPole-v0 is solved with an average reward of 195, MountainCar-v0 with an average reward of -110, Pendulum-v1 maximizes the reward when balancing the pendulum upright, and Reacher-v4 is solved with an average reward of around -3.75.

We trained agents for 50k steps in all environments, except MountainCar-v0, where training was extended to 200k steps with actions repeated five times to enhance exploration. All PPO hyperparameters were kept in their default settings. In all experiments, unless stated otherwise, OSWM was provided with 1k context steps collected from the real environment using random actions.

### 4.1.1 QUANTITATIVE EVALUATION OF AGENT PERFORMANCE

In Table 1, we compare the average performance of 100 test episodes for three agents: OSWM-PPO, PPO, and a Random Baseline. OSWM-PPO is trained purely on the dynamics predicted by OSWM using 1k context steps from the real environment, while PPO is trained only on the real environment, and the Random Baseline selects actions randomly. Since OSWM's synthetic rewards may not be indicative of the current agent's performance on the real environment, we evaluate each agent periodically after 100 training steps. Moreover, as discussed below in Section 4.1.2, training the agent too long on OSWM can result in performance degradation. Therefore, we apply an early stopping heuristic that takes the best agent training checkpoint. We do this on a per-seed-basis and compute the mean over multiple seeds.

In GridWorld, OSWM-PPO matches PPO with a reward of 5.2, outperforming the random baseline (-14.2) and demonstrating robustness in simple environments. In CartPole-v0, OSWM-PPO achieves 196.5, close to PPO's 200 (random baseline: 21.3). Also in SimpleEnv, OSWM-PPO reaches -4.7, performing well compared to PPO (-0.8) and significantly better than the random baseline (-256.2). These results are particularly surprising, as they show that pretraining on synthetic dynamics generated by random, untrained neural networks can still lead to strong performance in certain tasks, even without direct training on real environment data.

| Environment | OSWM-PPO | PPO | Random Baseline |
|---|---|---|---|
| GridWorld | 5.2 ± 0.0 | 5.2 ± 0.0 | -14.2 ± 0.3 |
| CartPole-v0 | 196.5 ± 4.2 | 200.0 ± 0.0 | 21.3 ± 3.9 |
| SimpleEnv | -4.7 ± 5.2 | -0.8 ± 0.1 | -256.2 ± 16.6 |
| MountainCar-v0 | -200.0 ± 0.0 | -110.5 ± 2.1 | -200.0 ± 0.0 |
| Pendulum-v1 | -1185.4 ± 31.2 | -268.9 ± 22.2 | -1230.3 ± 8.6 |
| Reacher-v4 | -10.2 ± 0.9 | -4.6 ± 0.3 | -42.8 ± 0.3 |

Table 1: Average performances over 3 seeds on 100 test episodes of 3 different agents (higher values are better). OSWM-PPO is a PPO agent trained only on the OSWM, PPO is a PPO agent trained on the real environment and the random baseline is an agent taking random actions. All agents are evaluated on the real environment, and we apply an early stopping heuristic for each seed before we compute the mean.

In more complex environments like MountainCar-v0 and Pendulum-v1, OSWM-PPO struggles to match PPO, with larger gaps in rewards, indicating that the approach here is less effective. However, for Reacher-v4, OSWM-PPO shows noticeable improvement, coming closer to PPO performance and performing far better than the random baseline. In MountainCar-v0, the model appears inferior at interpolating behavior in unseen states or areas of the environment, as the random context covers only a small part of this task. In contrast, Pendulum-v1 should benefit from better exploration through random actions, as the initial state covers all pendulum rotations, and the random actions provide a wide range of velocities. Despite this, OSWM does not provide sufficiently accurate dynamics to support effective training, suggesting that Pendulum-v1 requires more precise control and dynamic predictions than OSWM can currently offer. This may be due to the inherent difficulty posed by these environments, including sparse rewards and continuous action spaces, which likely require more sophisticated priors to improve performance.

### 4.1.2 PERFORMANCE PROGRESSION ACROSS TRAINING STEPS

To better understand the progression of agent training when training on OSWM, we report the learning curves in Figure 2. Here, we depict the mean evaluation rewards over training steps for three PPO agents using OSWM, with the best and worst performances highlighted.[1] Performance is measured on the real environment over 10 test episodes.

In the GridWorld environment (left), agents quickly solve the task after minimal interaction, with only one agent showing slightly suboptimal behavior after about 15,000 steps. This demonstrates the robustness of OSWM in simple environments.

For CartPole-v0 (center), agents show strong early performance, with the mean curve stabilizing after a brief drop. The best-performing agent continues to improve, while the worst-performing agent experiences a notable drop-off later in training. This phenomenon, where initial improvements are followed by a decline, can be attributed to gaps in the OSWM's understanding of certain environment dynamics. For instance, OSWM might model the dynamics accurately at higher angular velocities but struggle at lower velocities, failing to account for subtle drifts that are not captured. As a result, the agent may receive overconfident reward signals, leading to poor performance when these unmodeled drifts become significant in the real environment.

In SimpleEnv (right), agents exhibit a sharp initial increase in performance, followed by a plateau or decline. The worst-performing agent's reward nearly returns to its initial level, highlighting variability in training outcomes. This suggests that while OSWM supports learning, the one-shot prediction approach can introduce variability in performance, particularly in continuous environments where fine control is crucial.

### 4.2 STUDYING THE PRIOR

In this section, we analyze the behavior of the Neural Network (NN) prior used in OSWM, which generates diverse dynamics through randomly initialized neural networks. To understand the state

---

[1]We point out that the mean curves in Fig. 2 do not use the early stopping heuristic and therefore, do not correspond to the mean values of Tab. 1 where we take the mean over early stopped agents.

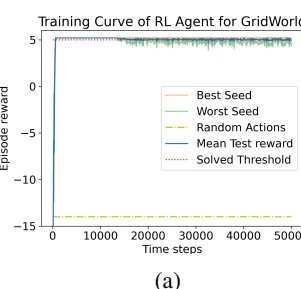 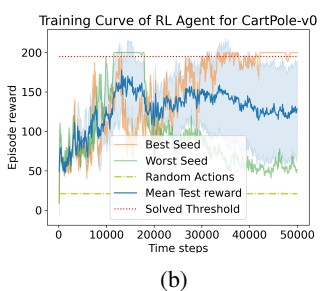 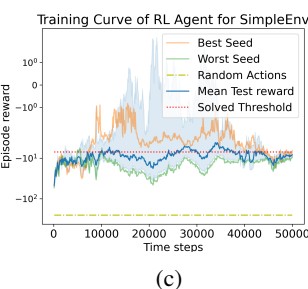

|     |     |     |
| :-: | :-: | :-: |
| (a) | (b) | (c) |

Figure 2: Evaluation scores for RL agent training on the OSWM for GridWorld, CartPole-v0, and SimpleEnv. Blue shows the mean over 3 runs, with the standard deviation in light blue. Orange and green depict the best and worst-performing agents, respectively.

dynamics produced by the NN prior, we sample batches of data, reflecting what OSWM encounters during training. For each prior dimension (e.g., the agent's position in GridWorld), we calculate the minimum and maximum values and divide them into 100 equal bins, visualizing the distribution for each dimension. The histograms in Fig. 3 show three distinct types of distributions produced by the NN prior. Some prior dimensions exhibit highly peaked distributions, as shown in Fig. 3a, where most values fall within a narrow range. For other dimensions, we observe broader distributions with a more even spread of values, as seen in Fig. 3b. Finally, some prior dimensions follow multimodal distributions, with two or more distinct peaks, as depicted in Fig. 3c. This pattern of three distinct distribution types is commonly observed across various dimensions.

The variation in distribution types suggests that the NN prior can capture both simple, deterministic behaviors and more complex, multimodal scenarios. However, as shown in Table 2 (left), using only the NN prior impacts OSWM-PPO performance in environments like CartPole-v0, where momentum is key for modeling the pole's angular dynamics. In contrast, GridWorld and SimpleEnv, which do not entail momentum, perform similarly to when both the NN and momentum priors are used (see Table 1). MountainCar-v0, Reacher-v4, and Pendulum-v1 were unsolvable before, and as expected, removing complexity from the prior does not make them solvable. This highlights that while the NN prior's multimodality supports diverse behaviors, it is insufficient for tasks that rely on accurate momentum-based dynamics. The distributions of the momentum prior are reported in Appendix A. The right column of Table 2 on improved context generation is analyzed further in Section 4.3.

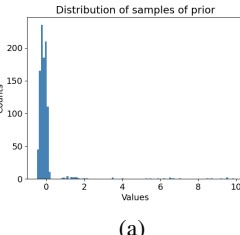 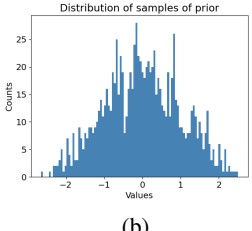 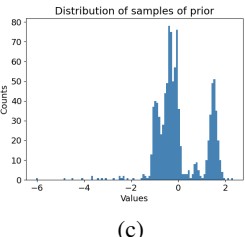

|     |     |     |
| :-: | :-: | :-: |
| (a) | (b) | (c) |

Figure 3: Typical distribution patterns generated by the NN prior: (a) highly peaked, (b) broad, and (c) multi-modal distributions.

### 4.3 STUDYING THE CONTEXT SAMPLING

Context is crucial for the predictive performance of OSWM. This section explores how different context sampling methods affect the model's predictions. Assessing the role of sampling strategies requires multiple agent trainings in OSWM and evaluations across multiple test episodes and environments. Since this is computationally expensive, we make use of a proxy dataset to evaluate the effectiveness of various sampling strategies more efficiently. The details of the generation of the proxy set are provided in Appendix B, but a high-level overview is given here.

The proxy set is created from transitions collected in the real environment using a PPO agent trained to perform at the expert level. First, the PPO agent is used to generate 5000 expert transitions across

| Environment | NN Prior Only | Improved Context |
|---|---|---|
| **GridWorld** | $5.2 \pm 0.0$ | $3.9 \pm 1.9$ |
| **CartPole-v0** | $191.7 \pm 11.2$ | $108.0 \pm 34.6$ |
| **SimpleEnv** | $-1.3 \pm 0.3$ | $-2.5 \pm 0.8$ |
| **MountainCar-v0** | $-200.0 \pm 0.0$ | $-200.0 \pm 0.0$ |
| **Pendulum-v1** | $-1217.4 \pm 40.9$ | $-1245.0 \pm 25.1$ |
| **Reacher-v4** | $-10.0 \pm 0.6$ | - |

Table 2: Average performances when the OSWM is trained with the NN prior only (left; with randomly sampled context), as well as when a more sophisticated context sampling strategy is adopted (with NN+momentum prior). Higher values are better.

multiple episodes. From this, 500 transitions are sampled for each of three settings: 0% randomness (expert actions only), 50% randomness (half expert, half random) and 100% randomness (random actions only). This total of 1500 transitions spans expert behavior to exploratory actions and the proxy. The intuition behind mixing random and expert transitions is to cover states that are not typically encountered by an expert agent alone and thus, the proxy set can capture a wider range of environment dynamics. We then tested five different context sampling strategies: *random* (actions sampled uniformly), *repeat* (random actions repeated for three steps), *expert* (policy solving the environment), *p-expert* (mixing PPO expert and random actions 50/50), and *mixture* (first third random, second third *p-expert*, final third PPO expert).

For evaluation, OSWM is provided with 1000 context steps from each strategy, and the proxy set is used to assess their impact on model predictions (Table 3 in the appendix) by computing the mean squared error (MSE) between predicted dynamics and true targets from the proxy set. Based on the proxy loss, the best strategy is selected for each environment and evaluated in Table 2 (right). In complex tasks like MountainCar-v0 and Pendulum-v1 (using *mixture*), even with improved context, these environments remain unsolved. For Reacher-v4 (*random*), simple random sampling proves best, reflecting that basic methods can sometimes capture the necessary dynamics. In SimpleEnv (*p-expert*), the improved context sampling enhances performance. GridWorld (*mixture*) sees minimal variation, with random sampling generally being sufficient to capture its simpler dynamics. Overall, *p-expert* and *mixture* often yield the best results, while *repeat* and *expert* strategies are less effective. *Random* proves to be a reliable default, offering solid performance across many environments.

## 5 CONCLUSION

We introduced One-Shot World Model (OSWM), a world model trained purely on synthetic data sampled from a prior distribution based on randomly initialized, untrained neural networks by leveraging In-context Learning. Despite the simplicity of the prior, OSWM achieved promising results as it is able to train RL agents to solve tasks like GridWorld and control tasks such as CartPole-v0, demonstrating the potential of synthetic pretraining in Model-Based Reinforcement Learning. Although the model still struggles with more complex environments like Pendulum-v1 and MountainCar-v0, our empirical analysis suggests that improving the priors and refining context sampling are key to enhancing performance. Our results highlight the potential of synthetic pretraining in RL, suggesting that with further optimization, this approach could be a key step towards foundation world models, capable of tackling increasingly complex tasks. With further optimization of the prior, synthetic pretraining could enable the development of more generalizable foundation world models, offering a scalable solution for RL training, especially when evaluating real-world environments is costly and challenging.

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

## A STUDYING THE MOMENTUM PRIOR

Unlike the Neural Network prior, the Momentum prior is based on physics-driven dynamics, modeling velocity and positional updates to simulate environments with simple physical laws.

To analyze the behavior of the Momentum prior, we generate histograms in the same manner as with the NN prior, sampling batches of data and calculating the minimum and maximum values for each dimension. These dimensions reflect aspects like velocity and position, which are updated according to basic physical interactions such as gravity or action forces. The range of each dimension is then divided into 100 equal bins, and the occurrences in each bin are counted to visualize the distribution of values.

The Momentum prior produces a variety of distributions across dimensions, as shown in Figure 4. In some cases, we observe broad distributions with values spread uniformly across the range (Fig. 4a). This often occurs in environments with elastic reflections or angular motion without gravity. In other cases, the distribution is multi-modal, featuring multiple peaks (Fig. 4b), which can arise from non-elastic reflections or angular dynamics with insufficient torque to overcome gravity. Finally, some dimensions exhibit sparse distributions (Fig. 4c), where values cluster into a few discrete states. This pattern typically results from environments lacking friction or other forces that would normally smooth out the motion.

These distribution patterns reflect the diversity of physical interactions captured by the Momentum prior. Compared to the NN prior, the behavior here is more interpretable, as it directly corresponds to simplified physical models of motion and interaction.

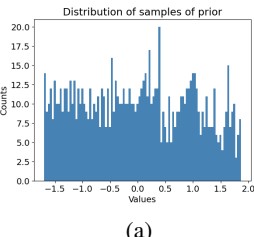 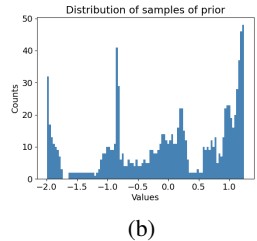 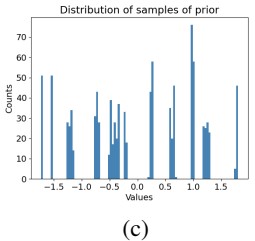

(a)                                   (b)                                   (c)

Figure 4: Typical distribution patterns generated by the Momentum prior: (a) broad, (b) multi-modal, and (c) sparse distributions.

## B CONTEXT GENERATION EVALUATION

To further investigate the effect of different context-generation strategies, we performed an evaluation using a proxy loss for full OSWM training and RL agent evaluation. The proxy set was constructed by generating transitions from a PPO agent trained on each specific environment. These transitions included state-action pairs, next states, and rewards.

We simulated different levels of randomness to capture a range of behaviors. Specifically, we generated rollouts with 0% randomness (only expert actions), 50% randomness (half expert, half random), and 100% randomness (only random actions). For each level of randomness, we collected 5000 transitions across multiple episodes and randomly subsampled 500 transitions per level, resulting in a total of 1500 transitions per environment. This proxy set was used to compute the mean squared error (MSE) between the predicted dynamics from OSWM and the actual transitions.

The intuition behind why we believe the proxy set is effective lies in its ability to cover a wide range of environment dynamics. Certain environments, like MountainCar-v0, require exploration using both efficient, expert-like actions to solve the task, and suboptimal actions to discover diverse states in the environment. Similarly, for environments like CartPole, non-goal-oriented actions—such as those where the pole is not upright or the cart velocity is high—allow the model to observe critical states not typically encountered by an expert agent alone. By including random actions in the proxy set, we aim to capture these middle-ground dynamics, such as a scenario in MountainCar where a fast-moving car decelerates, a behavior not covered by either purely expert or random

actions. Additionally, this strategy helps represent the trajectory from suboptimal to successful actions, enhancing OSWM's capacity to generalize across different levels of agent performance.

The results for each context-generation strategy (random, repeat, expert, p-expert, and mixture) across the various environments are shown in Table 3. This table provides a detailed view of how the different strategies affect the proxy loss, which serves as a reliable proxy for predictive performance.

| Environment | Random | Repeat | Expert | p-expert | Mixture |
|---|---|---|---|---|---|
| GridWorld | 0.468 | 0.413 | NaN | 0.218 | **0.203** |
| CartPole-v0 | 0.0048 | 0.0054 | 0.0138 | **0.00079** | 0.00186 |
| MountainCar-v0 | 0.00065 | 0.00025 | 5e-05 | 1.19e-05 | **8.5e-06** |
| SimpleEnv | 0.38 | 0.701 | 9.614 | **0.103** | 0.139 |
| Pendulum-v1 | 0.025 | 0.03173 | 0.0779 | 0.0578 | **0.018** |
| Reacher-v4 | **0.312** | 0.552 | 1.456 | 0.347 | 0.322 |

Table 3: Proxy loss with respect to the different context generation techniques. Mean squared error loss for 1500 validation transitions in the corresponding environment. The best performance per environment is in bold.

## C  HYPERPARAMETERS

The hyperparameters for the OSWM can be found in table 4. For training the OSWM, the hyperparameters can be found in 5.

| Hyperparameter | Value |
|---|---|
| Embedding size | 512 |
| Number of Attention Heads | 4 |
| Hidden size | 1024 |
| Number of layers | 6 |
| Embedding size | 512 |

Table 4: Hyperparameters defining the OSWM transformer architecture.

| Hyperparameter | Value |
|---|---|
| Optimizer | AdamW |
| $\beta_1, \beta_2$ | $0.9, 0.999$ |
| $\epsilon$ | $1e^{-8}$ |
| Weight decay | 0.0 |
| Initial Learning Rate | $5e^{-5}$ |
| Batch size | 8 |
| Epochs | 50 |
| Steps per epoch | 100 |
| Warm-up epochs | 12 |
| Sequence length | 1001 |
| Maximum eval position | 1000 |
| Minimum eval position | 500 |
| Eval position sampling function | $p_i = \frac{1}{((\max - \min) - i)^q}$ |
| $q$ for eval function | 0.4 |

Table 5: Hyperparameters that define the training pipeline of the OSWM training.

## D  CUSTOM ENVIRONMENT DETAILS

This following section will describe the details of the custom environment used to evaluate the OSWM. First, the GridWorld environment will be described, afterward the SimpleEnv.

### D.1 CUSTOM GRIDWORLD

The GridWorld environment is designed as a simple environment with discrete states and actions. It is deliberately easy to solve, as its main goal is to test the modeling capabilities of the OSWM in an easy base case. This is further helped by the fact that for discrete spaces, the OSWM predictions are rounded to the next integer value. This allows us to use the same condition for termination. And give the agent the same interface as the real environment.

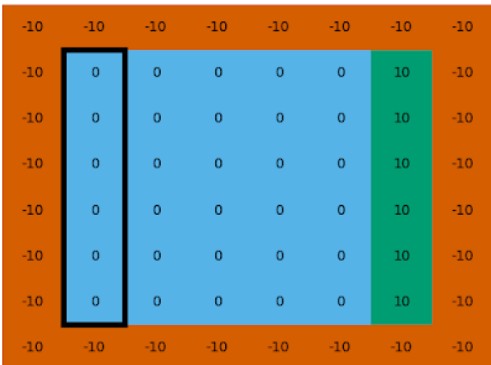

Figure 5: Visualization of the custom GridWorld environment. Terminal states are in red, goal states are in green, and initial states are highlighted in black. Immediate reward in the cells.

The GridWorld consists of an 8x8 grid. Observations are the x-postion and y-postion. Actions are 4 discrete moves (up, down, left, and right). With the outer ring of cells being terminal states with a negative ten reward. The goal states give a positive ten reward and are located in the second last column to the right. They span from the second row to the second last row. Each step gives a negative one reward and a small positive ($0.01 * x_{pos}$) for being further to the right. Episodes are truncated after exceeding 25 episode steps. The agent starts the episode at $x_{pos} = 1$ and with a $y_{pos}$ between 1 and 6. A visualization of the GridWorld can be found in fig. 5.

### D.2 CUSTOM SIMPLEENV

The SimpleEnv serves to provide a first intuition for continuous action and state space environments, while using simplistic dynamics. Similar to the GridWorld, it is designed to be easily solved by RL agents with smooth and dense goal-oriented rewards.

It has 1-dimensional continuous action space ($a \in [-10., 10.]$) and a 1-dimensional continuous state space ($s \in [-30., 30.]$). The immediate reward is the negative absolute state, $r = -1 \cdot abs(s)$. Episodes have a fixed length of 20 steps. The dynamics of the environment are defined by the action being added to the state, $s_t = s_{t-1} + a_t$. The initial state of the environment is sampled uniformly between -5 and 5.

### D.3 SOLVED REWARD FOR CUSTOM ENVIRONMENTS

To establish the solved threshold for custom environments, a relative score is determined based on a comparison between expert performance and random actions. This approach allows for the definition of a consistent threshold across various environments. The solved reward is calculated using the following equation:

$$R_{solved} = R_{max} - (R_{max} - R_{random}) \times 0.03 \tag{1}$$

In this equation, $R_{max}$ represents the expert-level performance, while $R_{random}$ is the expected reward when taking random actions. The coefficient of 0.03 is chosen as it aligns with the solved

threshold established for the CartPole-v0 environment, providing a standard for evaluating other environments.

# E  BO FOR PRIOR AND MODEL HYPERPARAMETER

In order to determine the ideal hyperparameter for both the OSWM model and the underlying prior, an automatic optimization was performed. For the prior, especially, the architecture of the neural networks in NN prior play a crucial role in its performance. The library used for this optimization is HpBandSter. The model is trained for a fixed 50 epochs, we omit using Hyperband, as it is unclear how the different complexity of priors plays into the reliance of the low-cost proxy for the OSWM. The optimization was performed for 45 iterations with 3 workers. The configuration space and results can be found in 6. The target function, being optimized, is the same validation loss used for evaluating the context generation types in Sec. 4.3.

| Hyperparameter | Type | Range/Choices | Final |
|---|---|---|---|
| Number hidden layer | integer | [1, 6] | 1 |
| Width hidden layers | integer | [8, 64] | 16 |
| Use bias | bool | [True, False] | False |
| Use dropout | bool | [True, False] | False |
| Dropout probability | cond. float | [0.1, 0.9] | - |
| Activation Functions | bool (each) | [relu, sin, sigmoid, tanh] | (sin, tanh) |
| Initial state scale | float | [1., 20.] | 18.14 |
| Initial state offset | float | [1., 5.] | 3.28 |
| Use layer norm | bool | [True, False] | True |
| Use residual connection | bool | [True, False] | True |

Table 6: Hyperparameters of the NN Prior optimized using BO. Each hyperparameter, with its type, the range or choices, and final best performing value.

For the optimization of the encoder and decoder models of the OSWM, the same optimization was performed. The baseline is a linear encoder and decoder, for more complex data, a more expressive encoder and decoder might aid in representation. Additionally, it allows us to separately encode action and state, and separately decode the next state and reward. The choices for encoding and decoding are a standard MLP or a model with separate MLPs concatenating both outputs, denoted with *Cat*. An overview of the entire configuration space and the results are given in table 7.

| Hyperparameter | Type | Range/Choices | Final |
|---|---|---|---|
| Encoder type | categoric | [MLP, Cat] | Cat |
| Encoder width | categoric | [16, 64, 256, 512] | 512 |
| Encoder depth | Integer | [1,6] | 3 |
| Encoder activation | categoric | [ReLU, sigmoid, GeLU] | GeLU |
| Encoder use bias | bool | [True, False] | True |
| Encoder use res connection | bool | [True, False] | True |
| Decoder type | categoric | [MLP, Cat] | Cat |
| Decoder width | categoric | [16, 64, 256, 512] | 64 |
| Decoder depth | Integer | [1,6] | 2 |
| Decoder activation | categoric | [ReLU, sigmoid, GeLU] | sigmoid |
| Decoder use bias | bool | [True, False] | False |
| Decoder use res connection | bool | [True, False] | True |

Table 7: The hyperparameters of the encoder and decoder of the OSWM optimized using BO. Each hyperparameter, with its type, the range or choices, and final best-performing value.

