# OpenReview forum: "One-shot World Models Using a Transformer Trained on a Synthetic Prior"
_ICLR.cc/2025/Conference — ICLR 2025 Conference Withdrawn Submission_

### Official Review · Reviewer_X52E · 2024-10-27

**Soundness:** 3
**Presentation:** 1
**Contribution:** 2
**Rating:** 3
**Confidence:** 4

**Summary:**

The paper explores synthetic pertaining by generating samples from randomly initialized neural networks and learning world models with it. Using the learnt world model, the authors evaluate performances in selected RL tasks such as CartPole, Reacher, MountainCar, Pendulum, etc.

**Strengths:**

- The idea is sound.
- I also like that the authors kept the randomly initialized NN simple with recurrent units.

**Weaknesses:**

Overall, the paper feels incomplete, with multiple concerns as stated below.

- Writing: Section 3.2 was difficult to go through. Besides, I would encourage the authors to improve the captions on the Figures.
- Techniques: The momentum prior feels engineered to the task at hand. Ideally, I'd like to see how this prior (or an update of this) can help in a wide range of tasks.
- Experiments: The authors test their work on a small subset of the environments. Given the work introduces synthetic samples, I'd like to see experiments in a more established benchmark such as Atari100k and/or DMC.

**Questions:**

- In case the termination condition is not known, how can you learn the termination conditions in the context of your current work?
- Line 197: action $a_t$ = [...] was not clear. Is there a typo?
- What is the motivation for restricting velocity between -3 and 3?
- How do you randomly initialize the NN?

---

### Official Review · Reviewer_nPiE · 2024-11-02

**Soundness:** 2
**Presentation:** 2
**Contribution:** 1
**Rating:** 3
**Confidence:** 4

**Summary:**

This work proposes pre-training a transformer world model on synthetic data generated from untrained neural networks and physics priors. Then it proposes using it as a world model for RL to one-shot solve new tasks. The approach is empirically evaluated in simple tasks, capable of solving some of them with only 1000 transition samples.

**Strengths:**

1. I find the proposed approach simple and novel

**Weaknesses:**

1. In my opinion this paper falls more on the empirical contributions spectrum. Through that lens I find the presented results too limited and not sufficiently impactful. At this stage, the work shows signs of life but needs more convincing results on more difficult tasks in order to have the impact on the scientific community. For example, there has been world model work which assumes certain properties in the deepmind control suite [Hao et al, 2021](https://arxiv.org/abs/2112.02817) and gets superb sample efficiency and similar examples in manipulation such as [Biza et al., 2022](https://arxiv.org/pdf/2202.05333). I would suggest aiming for that level of task complexity.
2. For the proposed method to work sufficiently well, it requires context data from a distribution of policies with different performances. This is a fundamental limitation which makes the application of this approach to true RL problems very limited. However, I do acknowledge that it is still applicable to low-data regimes such as imitation learning.
3. The paper will benefit from a more concise presentation. Certain sections, particularly 3, 4.1, could be streamlined to emphasize key findings and enhance readability. A tighter structure might help maintain reader engagement and highlight the study’s core contributions more effectively. If you intend to keep the paper content as is, I would recommend fitting it within 6 pages (which is perfectly acceptable this year).
4. While the paper idea is novel and unique, I feel that the baselines and evaluations could be improved. Currently the paper uses PPO as the upper bound and a random policy as the lower bound in Table 1. Then it simply poses the question, where does our method stand but doesn't really compare properties of the algorithm. For example, one could compare sample efficiency vs standard baselines such as Dreamer or very sample efficient world models such as ED2  [Hao et al, 2021](https://arxiv.org/abs/2112.02817). On the other hand, since the proposed method is using data from a mixture of policies, it would also be worthwhile to compare against offline trained world models such as TDMPC2 [(Hansen et al. 2024)](https://arxiv.org/abs/2310.16828) or PWM [(Georgiev et al. 2024)](https://arxiv.org/abs/2407.02466).
5. The experiment in Section 4.2 provides an unfair comparison as it changes the type of context in Table 2, effectively selecting the best hyper-parameters per task. Doing this limits the generalizability of your approach and makes it unclear what context selecting strategy is the best.

Minor commnets:
1. Line 039 states that transformers have pushed the boundaries of world models. However, at least in the cited RL applications, I (and many colleagues) find that transformers have not shown sufficient improvements over simpler models such as MLPs. As such I found this claim false and misleading to readers.
2. I find the first paragraph of your related work to be unnecessarily long. I would recommend only referencing work that (1) shows a need for learning better world models or (2) is directly relevant to yours.
3. The ablation in Section 4.2 is very interesting. I'd encourage you to bring Table 3 into to main text. Additionally, 'Improved Context' is difficult to understand from the text
4. Some figures are difficult to read. As a rule of thumb, try to have text in figures to be roughly the same size as your main text.

**Questions:**

1. Learning entirely from synthetic data is an interesting idea and problem constraint but have you also considered applications of your approach to extremely limited data regimes? For example, robotics.
2. Related to the above, what about cases where we progressively obtain more data. Have you explored the direction of fine-tuning your world model as you interact with the environment similar to a standard MBRL algorithm.
3. The decision to use a full transformer encoder-decoder is a bit uncommon and unmotivated. Why did you decide on that?

---

### Official Review · Reviewer_Ch12 · 2024-11-04

**Soundness:** 2
**Presentation:** 2
**Contribution:** 3
**Rating:** 5
**Confidence:** 4

**Summary:**

This paper introduces the One-Shot World Model (OSWM), a transformer-based model that learns a world model solely from synthetic data rather than real-world observations. Leveraging randomly initialized neural networks as a prior, OSWM generates diverse environment dynamics, which allows it to train a world model. At inference, the model is provided 1000 transitions from the target environment and the RL policy is trained entirely in the OSWM generated data. The model achieves promising results in simple environments, such as GridWorld and CartPole, where it adapts effectively using only 1,000 transition samples as context.

**Strengths:**

- The problem studied here is novel and quite interesting.
- The results, while not very exceptional, are still promising and show the potential of synthetic data for real problems.

**Weaknesses:**

- I wonder if it is right to call the model a one-shot model. During inference the model uses 1000 transitions as in-context examples which may comprised different number of episodes depending on the environment. In some cases, the environment maybe non-episodic. In general 1-shot refers to using 1 in-context examples. I believe calling 1000 transitions as one in-context example can be a bit misleading. Maybe a more general term would be few-shot or in-context world models?
- A more thorough analysis of the strength of the world model and its implications on the downstream tasks would be nice to have for the paper. For example, in table 3 the authors compare various sampling strategies for in-context learning and then select the best one (based on mse) for reporting results in table 2 (left). However it is not clear whether better mse on the 1500 held out transitions does always lead to better downstream RL performance? It would be nice to have an analysis comparing the world model performance and the corresponding downstream RL performance.
- It seems that for each dimension of the state space, a different neural network is used. This seems very limiting as the different dimensions of the state space in the synthetic data may lack global consistency and may thus affect rl performance. I wonder if the authors have tried more sophisticated networks that produce the whole state using a single network. Also I wonder if the authors tried using a recurrent network for generating consecutive states in the synthetic data?
- I believe one baseline that would be good to have is what if you train a world model on the 1000 in-context transitions used during inference and then learn an RL policy based on that world model. This baseline would shed more light on the effectiveness of synthetic data.

**Questions:**

-  How much synthetic data is used for training the world models? How does the downstream RL performance scale with more synthetic data.

---

### Official Review · Reviewer_v5Tr · 2024-11-06

**Soundness:** 4
**Presentation:** 4
**Contribution:** 3
**Rating:** 6
**Confidence:** 3

**Summary:**

This paper proposes One-Shot World Model (OSWM), a transformer-based world model trained entirely on synthetic data. Unlike traditional world models trained on environment observations, OSWM is pretrained entirely on synthetic data generated from a prior distribution composed of randomly initialised neural networks. OSWM can then be used as a world model for a given environment by leveraging in-context learning. Experiments demonstrate OSWM's ability to train RL agents that solve simple environments like GridWorld, CartPole, and a custom control task. However, the model struggles with more complex environments.

**Strengths:**

- Training world models entirely on synthetic data generated from randomly initialized neural networks is a novel and intriguing idea.
- Promising results on simple environments: The authors demonstrate successful agent training on simple environments purely from synthetic priors. It suggests the potential of this approach for rapid adaptation to new tasks. Leveraging in-context learning allows for quick adaptation to unseen environments without extensive retraining.
- The paper provides a detailed analysis of the impact of different prior components, specifically analyzing the distribution of states produced by the NN prior and the momentum prior. They also study the impact of different context sampling strategies.
- The paper provides a promising first step for further research in pretraining of RL world models with synthetic data.

**Weaknesses:**

- Limited applicability to complex environments: The current model struggles with harder environments highlighting the need for further development.

**Questions:**

- To study the effectiveness of synthetic pretraining, I think it would be useful to have a baseline world model that is trained from scratch using the in context samples provided to OSWM.
- I am a bit unclear on how the momentum prior is used in the prior training. Section 3.2 mentions the NN prior and the momentum prior are concatenated. But Fig 1 shows that the entire state vector $x_t = [s_t^{1:d_s}, a_t^{1:d_a}$ so $x_t$ is just the NN prior, so where is the momentum prior concatenated? I do not see $v_t$ or $p_t$ in Fig 1.

---

### Note · Authors · 2024-11-18

I have read and agree with the venue's withdrawal policy on behalf of myself and my co-authors.